# A Clinical Decision Web to Predict ICU Admission or Death for Patients Hospitalised with COVID-19 Using Machine Learning Algorithms

**DOI:** 10.3390/ijerph18168677

**Published:** 2021-08-17

**Authors:** Rocío Aznar-Gimeno, Luis M. Esteban, Gorka Labata-Lezaun, Rafael del-Hoyo-Alonso, David Abadia-Gallego, J. Ramón Paño-Pardo, M. José Esquillor-Rodrigo, Ángel Lanas, M. Trinidad Serrano

**Affiliations:** 1Department of Big Data and Cognitive Systems, Instituto Tecnológico de Aragón, ITAINNOVA, María de Luna 7-8, 50018 Zaragoza, Spain; glabata@itainnova.es (G.L.-L.); dabadia@itainnova.es (D.A.-G.); 2Escuela Universitaria Politécnica de La Almunia, Universidad de Zaragoza, Calle Mayor, 5, 50100 La Almunia de Doña Godina, Spain; lmeste@unizar.es; 3Infectious Disease Department, University Clinic Hospital Lozano Blesa, San Juan Bosco 15, 50009 Zaragoza, Spain; joserrapa@gmail.com; 4Department of Medicine, Psychiatry and Dermatology, University of Zaragoza, 50009 Zaragoza, Spain; alanas@unizar.es (Á.L.); tserrano.aullo@gmail.com (M.T.S.); 5Aragon Health Research Institute (IIS Aragon), 50009 Zaragoza, Spain; 6Internal Medicine Department, University Clinic Hospital Lozano Blesa, San Juan Bosco 15, 50009 Zaragoza, Spain; mjesquillor@gmail.com; 7Digestive Diseases Department, University Clinic Hospital Lozano Blesa, San Juan Bosco 15, 50009 Zaragoza, Spain; 8CIBEREHD, 28029 Madrid, Spain

**Keywords:** COVID-19, ICU, mortality, machine learning, predictive model, clinical decision web tool

## Abstract

The purpose of the study was to build a predictive model for estimating the risk of ICU admission or mortality among patients hospitalized with COVID-19 and provide a user-friendly tool to assist clinicians in the decision-making process. The study cohort comprised 3623 patients with confirmed COVID-19 who were hospitalized in the SALUD hospital network of Aragon (Spain), which includes 23 hospitals, between February 2020 and January 2021, a period that includes several pandemic waves. Up to 165 variables were analysed, including demographics, comorbidity, chronic drugs, vital signs, and laboratory data. To build the predictive models, different techniques and machine learning (ML) algorithms were explored: multilayer perceptron, random forest, and extreme gradient boosting (XGBoost). A reduction dimensionality procedure was used to minimize the features to 20, ensuring feasible use of the tool in practice. Our model was validated both internally and externally. We also assessed its calibration and provide an analysis of the optimal cut-off points depending on the metric to be optimized. The best performing algorithm was XGBoost. The final model achieved good discrimination for the external validation set (AUC = 0.821, 95% CI 0.787–0.854) and accurate calibration (slope = 1, intercept = −0.12). A cut-off of 0.4 provides a sensitivity and specificity of 0.71 and 0.78, respectively. In conclusion, we built a risk prediction model from a large amount of data from several pandemic waves, which had good calibration and discrimination ability. We also created a user-friendly web application that can aid rapid decision-making in clinical practice.

## 1. Introduction

### 1.1. Context

Recent advances in the field of artificial intelligence have demonstrated their success in different fields of interest such as the environment [1], climate change [2], agriculture [3], industry [4] and health [5,6,7], among others. In particular, the application of modelling and the development of machine learning algorithms have been emerging in recent times and have been responsible for these multiple applications of interest for data-driven decision support and profit maximisation.

Deep Learning is a subset of machine learning algorithms that has shown great promise especially in the application of data with some temporality [8,9], image or textual data in areas such as image recognition [10,11,12,13], natural language processing [14] or speech recognition [15].

These algorithms generally outperform machine learning techniques when the data size is large. This explains the success in these areas where large amounts of information are generated, such as time series data retrieved by sensors in real time or images or textual knowledge generated through the Internet. In addition to this, the success is also partly due to the fact that deep learning algorithms allow automatic feature extraction based on the data, unlike shallow machine learning algorithms whose features must be previously identified by a subject matter expert.

Deep learning algorithms and in particular deep neural networks allow through their layered architecture to extract from more general to more specific features, without the need for domain expertise. Therefore, when we have a large amount of information and/or unstructured data and/or domain knowledge is lacking, deep learning algorithms outperform others as there is no need to worry about feature engineering.

In addition to these advantages, deep learning techniques have the ability to adapt to different domains more easily [4], for example through transfer learning by using pre-trained deep neural networks.

However, as they are more complex techniques, training deep learning algorithms involve a higher computational cost and therefore requires more powerful infrastructure. Although it has been shown that Deep Learning sometimes outperforms classical machine learning techniques, when the data size is not too large, the latter are preferable [16].

In addition to having a lower computational cost and being recommended for smaller datasets, another advantage of shallow machine learning algorithms is that they are easier to interpret and understand compared to Deep Learning where deep networks are trained as a “black box”. This interpretability may be the main argument why many sectors use machine learning techniques as opposed to Deep Learning. This may be the case in the field of health and medicine, when the aim is to provide tools to clinicians helping them make critical decisions about a patient based on information about the patient at a specific time, which is the scope of the work we present here.

The application of artificial intelligence techniques in the field of health and medicine has been particularly accentuated in recent months to create problem-solving strategies due to the COVID-19 (Coronavirus disease 2019) pandemic [17].

The COVID-19 pandemic has already affected more than 174 million people, causing nearly 3 million deaths and overloading healthcare systems worldwide [18]. The clinical severity of the disease is highly variable. Most cases are asymptomatic or mild, but 14% of patients have severe disease, and 5% are critical [19]. The case fatality rate in China has been reported to be 2.3%, and in Europe it reaches 4–4.5% [20].

Multiple predictive and prognostic factors associated with increased severity of infection have been described, and models have been built to assist healthcare systems prioritize medical attention [21,22,23]. Nevertheless, most of these models have significant biases [23]. In addition, the sample sizes have been limited and most models lack external validation or calibration. There have also been several pandemic waves, in which both mortality rate and severity have differed [24], so it would be necessary for models to prove useful in different scenarios over time.

Machine learning has become an advanced tool for diagnosing health problems and predicting the severity and mortality of different diseases. During the COVID-19 pandemic, machine learning tools have been used for a variety of investigations, ranging from image analysis for diagnosis of SARS-CoV-2 pneumonia to predicting future pandemic waves [17]. These algorithms have the advantage of being able to combine a large amount of information, extracting the most predictive characteristics of the diagnosis associated with coronavirus disease [17,21,22,23,25,26,27,28,29,30,31].

Therefore, during the COVID-19 pandemic, Machine Learning has been crucial in developing tools for a variety of investigations, ranging from image analysis for diagnosing SARS-CoV-2 (Severe Acute Respiratory Syndrome Coronavirus 2) pneumonia, predicting future pandemic waves, detecting multiple predictive and prognostic factors associated with increased severity of infection, to building severity models that help health systems prioritise care [8,21,22,23,24,25,26,27,28,29,30].

In the present study, we focus on assessing the prediction of intensive care unit (ICU) admission or mortality risk (severity risk) among patients admitted to the hospital with COVID-19 using machine learning algorithms.

### 1.2. Related Work and Limitations

Any predictive model must contemplate certain good practices for model building and validation that provide generalizability and easy interpretation. The first thing that must be clearly defined is the target population, as well as the prediction horizon (outcome), so that the model can be validated with good reliability. A lack of or incorrect definition can lead to inappropriate use of the model and biased interpretation of the results. Limited sample sizes are also a known problem when building prediction models, which also implies a high risk of bias and overfitting of the model [32].

Several studies have built predictive models and analysed mortality risk in patients with suspected or confirmed COVID-19, but with an unclear definition of the target population, without specifying the outcome (mortality) period, and at high risk of biases [33,34,35,36]. This bias can lead to miscalibrations and overestimation of the discrimination performance, especially when they are not externally validated.

Yan et al. [34] (mortality risk in validated or suspected COVID-19 inpatients) and Shi et al. [36] (death or severe COVID-19 in inpatients with confirmed COVID-19 at admission) validated their models with samples of patients with a different severity than the training set, which can lead to bad calibration. In particular, Yan et al. used temporal validation, selecting only severe cases and Shi et al. validated using less severe cases.

Yue et al. [37] validated their prognostic models (hospital stay of more than 10 days per COVID admission) using 5-fold cross-validation. Although this type of validation is adequate (cross-validation) in order to avoid the overfitting, external validation ensures better generalization of the model, though it is a more difficult type of validity to achieve. Gong et al. [38] (severe COVID-19 infection within minimum 15 days in inpatients with confirmed COVID-19 at admission) and Xie et al. [39] (mortality in hospital in inpatients with confirmed COVID-19 at admission) performed external validation and achieved good performance, though their validation was performed for different centres, possibly due to being early prognostic model studies of the pandemic (with a population from China); the populations studied were patients admitted between January and March 2020 (first pandemic wave). However, subsequent studies [24] have shown that there were differences between the pandemic waves in terms of patient characteristics and severity. In addition to model discrimination, calibration should also be assessed, though this is not always done or not done correctly. Xie et al. [39] (in-hospital mortality of inpatients with confirmed COVID-19 at admission) performed validation and assessed the calibration correctly.

Many recent studies have explored machine learning techniques for the generation of a COVID-19 prognostic model to predict disease severity, demonstrating their great potential [17,25,26,27] and the ability of these techniques to select the most significant/influential variables [28,29,30]. Variable selection can be crucial for the generation of a prognostic model in such a medical and emergency setting. It is well known good practice to select the simplest possible model that still achieves an acceptable level of performance [40]. This is even more important in the medical domain, where a limited number of observations [23] but a large number of variables may be available, which may lead to overfitting. Finally, COVID-19 prognostic models of disease severity ultimately aim to provide a user-friendly tool for clinical practice that can aid rapid decision-making [22,26,28]. This implies that, for its use to be feasible in practice, the model should not include a large number of variables to be introduced.

Yao et al. [27] built a model for detecting COVID-19 disease severity using the support vector machine (SVM) algorithm by finally selecting 28 features (clinical information and blood/urine test data). They used a dataset of 137 patients hospitalized between January and February 2020. Marcos et al. [28] explored several machine learning algorithms (regularized logistic regression, random forest, XGBoost) to identify early patients who will die or require mechanical ventilation during hospitalization. For training and internal validation, they used a cohort of 918 confirmed COVID-19 patients admitted to the Salamanca hospital between March and April 2020 and a cohort of 252 COVID patients from another hospital (admitted to the Barcelona clinical hospital between February and April 2020). To develop a user-friendly and practical calculator, the number of features used by the machine learning model was reduced from 140 to fewer than 10 (demographic variables, comorbidities, chronic medical treatment, clinical characteristics, physical examination parameters, and biochemical parameters). With a similar aim, Patel et al. [30] developed models for predicting the need for intensive care and mechanical ventilation using a cohort of 212 patients. To this end, they considered various machine learning techniques (random forest, MLP, support vector machines, gradient boosting, extra tree classifier, adaboost). Knight et al. [22] developed and validated a pragmatic risk score (4C mortality score) based on eight variables (age, sex, number of comorbidities, respiratory rate, peripheral oxygen saturation, level of consciousness, urea level, and C-reactive protein) with the aim of predicting mortality in patients admitted to the hospital with COVID-19. For this purpose, they used a large patient dataset including patients from 260 hospitals. In particular, model training was performed on a cohort of patients recruited between 6 February and 20 May 2020, and validation was performed on a second cohort of patients recruited between 21 May and 29 June 2020.

### 1.3. Contributions of Our Work

The aim of our study was to build a model for predicting the risk of ICU or death in hospitalised patients with COVID-19 and integrate it into an easy-to-use web application to aid rapid decision-making in clinical practice, which is very important in these times of pandemic. To this end, machine learning techniques were applied following a good model building and validation exercise.

As described in the previous section, some related studies have some limitations or shortcomings related to different aspects related to the definition of the study population, the sample size, the validation and generalisation of the model, the assessment of the calibration of the model or its practical application with the development of a web application for decision making. Our work addresses all these limitations.

Our model was built with a larger sample size than most of the studies reviewed, covering more temporal information, including several pandemic waves. The model was validated both internally and externally and its calibration was properly assessed. In addition to a correct validation of the model and good model performance values, we developed a user-friendly tool for quick decision making, which is ultimately the real goal in practice. In order to make it feasible to use in practice, the most important input variables were selected.

Specifically, starting from a set of more than 150 clinical variables based on the patient′s history and analysis carried out at hospital admission, we built a predictive model using machine learning algorithms with a reasonable dimensionality, including the 20 most predictive variables, which makes it easy to apply from a clinical perspective, that were also validated in a subsequent wave. In addition, we provide a user-friendly tool for practical use.

The following sections present the data retrieval and pre-processing process of the patient information, the statistical techniques and the process of generation and validation of the model carried out. Subsequently, the results are shown and finally the results obtained are discussed and compared with the state of the art and the main conclusions of our work are drawn.

## 2. Materials and Methods

### 2.1. Patient Information Recruitment

This study involved data from patients with SARS-CoV-2 infections confirmed by RT-PCR (Reverse Transcription Polymerase Chain Reaction) who were hospitalized in the SALUD hospital network of Aragon (Spain), which comprises 23 hospitals, between February 2020 and January 2021. The selection criterion was hospitalization within the first 20 days after, and no more than 10 days before, the first positive SARS-CoV-2 PCR test.

Patient information was retrieved through the BIGAN Gestión Clínica platform of the Department of Health of Aragon. This platform accesses the Aragon Health Records Database, which is the primary data source of SALUD, containing demographic and clinical information on patients. The original database included 7498 patients and 165 variables. The outcome studied was ICU admission or mortality within 30 days of hospital admission, which we will refer to as an unfavourable outcome or severity. Each patient′s demographics, comorbidities, and medication prescribed in the 6 months prior to hospitalization were considered. Vital signs were recorded upon arrival at the emergency room. Laboratory variables were measured in the first 24 h. This laboratory information was only available for the largest two hospitals of the SALUD network, which accounted for more than 60% of all patients admitted with COVID-19 in the Aragon region during the study period.

To avoid bias due to missing patient information, which could erroneously and unrealistically influence the performance of the model, patients with <65% of the variables filled in were removed from the original database (listwise deleiton). Furthermore, outliers caused by possible human error in filling in the data were analysed and removed. After removing patients with more than 35% of the variables not filled in and outliers and erroneous data, the univariate mean imputation was carried out for the missing data (missing data were imputed as the mean value of the variable). We used imputation techniques with the purpose of a minimum loss in sample size, although we discarded multiple imputation regression methods due to the complexity of the data with a large number of variables.

Our analysis included 3623 patients.

### 2.2. Statistical Analysis

For model building, the pre-processed dataset with information from February 2020 to November 2020 was considered and split into three separate datasets as is usual in machine learning algorithms: 75% for the training model (fitting the model), 12.5% for model validation (selecting the best model configuration [hyperparameter set] from a set of candidates), and 12.5% for model testing (providing unbiased evaluation metrics that give a generalized value of the performance of the fitted model). In addition, a dataset including information from November 2020 to January 2021 was used for external validation of the final model in a different temporal scenario. We recruited the 626 cases distributed during the period shown in Figure 1 (green color) that verify the study inclusion criteria, these data are completely different to the generation model data and thus provide a validation on a different dataset not used for the building models.

Figure 1 shows the number of hospitalizations reported in the time period considered. Note that the model included information from two different waves of the pandemic, and the external validation included information from yet a different new wave.

For both datasets, we analysed descriptive variables by severity groups (unfavourable/favourable outcome). Medians and interquartile ranges were used for quantitative variables, and absolute and relative frequencies for categorical variables. Variables were compared between severity groups using Mann-Whitney and chi-squared tests as appropriate.

Figure 2 shows a flowchart of the data retrieval and preparation process that summarises the above.

### 2.3. Training and Validation

The final aim of the study was to implement a severity diagnostic model in a web tool/platform accessible by clinicians. Thus, by entering the necessary patient information, the probability of severity is provided. In practice, a user-friendly predictive tool such as this is limited by the amount of patient information (variables) entered; too much information is not feasible in real-world use. Therefore, it was necessary to apply techniques to reduce candidate variables.

Model building was performed in several steps. First, different machine learning algorithms were explored, including artificial neural networks (multilayer perceptron [MLP]), random forest, and gradient boosting trees. The gradient boosting tree algorithms, particularly extreme gradient boosting (XGBoost) [41], achieved the best performance in predicting unfavourable outcomes (ICU admission or death) and were chosen for model development. In addition, ensembles of decision tree methods have the advantage of providing estimates of the importance of variables from a trained predictive model. To report the model, the final importance of each variable is calculated as the average of the importance in each tree, which is calculated by the amount that each attribute split point improves the performance measure weighted by the number of observations for which the node is responsible [42]. This value provides a ranking of feature importance that was used for the selection and reduction of variables.

Second, a stepwise procedure was followed to reduce the number of variables (simpler models) so that the loss in performance was not significant. Models were initially trained using all 165 variables. After the best model with all variables was selected, i.e., the model with the highest discriminatory ability over the validation set, the importance of the variables was calculated and the 50 most important variables selected by means of the XGBoost algorithm. The process was repeated for the dataset using the 50 variables, selecting the 20 most significant variables. The final model was generated by using the dataset with 20 variables.

The discriminative ability of the models was assessed by the area under the receiver operating characteristics (ROC) curve (AUC). The 95% confidence intervals (CIs) were obtained using 2000 stratified bootstrap replicates. The tests for comparing the AUCs proposed by DeLong et al. [43] and Pepe et al. [44] were applied to assess the discriminatory difference between models. In the comparison between models, we considered as the best model the one that corresponds to the largest AUC value. The AUC values can be interpreted as 0.5 = this suggests no discrimination, so we might as well flip a coin. 0.5–0.7 = we consider this poor discrimination, not much better than a coin toss. 0.7–0.8 = acceptable discrimination. 0.8–0.9= excellent discrimination. >0.9 = Outstanding discrimination [45].

Regarding the model build, the model hyperparameters were optimized over a set of possibilities, choosing the best possible combination (best model) with the selected validation set in order to avoid overfitting. We implemented our model in the Optuna framework to achieve this goal. Optuna [46] is a define-by-run API that allows users to construct the parameter search space dynamically and implements both searching and pruning strategies. In our case, we used the Tree-structured Parzen Estimator (TPE) algorithm [47].

Figure 3 shows a flowchart summarises the above information representing the methodology carried out for the final model generation.

### 2.4. External Validation

An external dataset with information from November 2020 to January 2021 (external validation set) was used to validate the final model. The calibration (agreement between the probabilities predicted by the model and the real incidence) and discriminatory capacity of the model were analysed, as well as the performance of the model for each cut-off point through the following statistical metrics: accuracy (1), specificity (2), sensitivity (3), Youden (4), positive predictive value (PPV) (5), and negative predictive value (NPV) (6):(1)$$\mathrm{Accuracy}=\frac{\mathrm{TP}+\mathrm{TN}}{\mathrm{TP}+\mathrm{TN}+\mathrm{FP}+\mathrm{FN}}$$
(2)$$\mathrm{Specificity}=\frac{\mathrm{TN}}{\mathrm{TN}+\mathrm{FP}}$$
(3)$$\mathrm{Sensitivity}=\frac{\mathrm{TP}}{\mathrm{TP}+\mathrm{FN}}$$
(4)Youden index=Sensitivity+Specificity−1
(5)$$\mathrm{PPV}=\frac{\mathrm{TP}}{\mathrm{TP}+\mathrm{FP}}$$
(6)$$\mathrm{NPV}=\frac{\mathrm{TN}}{\mathrm{TN}+\mathrm{FN}}$$
where TP, TN, FP and FN are the number of true positives, true negatives, false positives and false negatives, respectively.

This clinical utility analysis provides the clinician with decision-making capability, offering a set of different possibilities for the optimal threshold depending on the criterion to be optimized, which may change over time and with circumstances.

The level of significance in the study was established at *p* < 0.05. Analyses were performed using R language programming v 4.0.3 [48] and Python language programming v 3.7.7. [49].

## 3. Results

The descriptive characteristics of the model generation and external validation datasets are provided in Table 1 and Table 2, respectively. Significant differences between groups based on the severity were observed for predictive variables in the model generation cohort, and for most of them in the validation cohort.

Regarding the modelling procedure, Table 3 contains the set of hyperparameters explored in training the different algorithms (MLP, random forest, and XGBoost), as well as their best combination of hyperparameters and the AUC achieved. In the case of MLP, the Early Stopping technique was applied to avoid overfitting. To enhance training, both classes were weighted to balance the training sample.

The ROC curves of the best models for each ML algorithm on the test dataset are shown in Figure 4. The AUCs obtained by the algorithms that combine decision trees (i.e., random forest and XGBoost) were superior to those obtained by the MLP, with XGBoost achieving the best performance (AUC = 0.8307). The curve comparison test showed a significant predictive improvement (*p* = 0.043) by considering the XGBoost model vs. MLP.

For the best algorithm (XGBoost), Table 4 shows the set of hyperparameters for which the best models were attained considering all variables, the 50 best variables, or the 20 best variables as candidate variables. Figure 5 is an importance diagram of the 20 variables that were finally selected.

Finally, the ROC curves for the final models analysing all variables, 50 variables, and 20 variables on the test dataset are shown in Figure 6. The AUC comparison revealed non-significant predictive improvement or loss (*p* > 0.1) when considering the best model with the 165 source variables (AUC = 0.8307; 95% CI 0.7856–0.8718), the best model with the 50 most important variables (AUC = 0.8138; 95% CI 0.7654–0.8571), and the best model with the 20 most important variables (AUC = 0.8153; 95% CI 0.7655–0.8615). Therefore, the most robust and parsimonious choice was to consider the 20-variable model as the final model due to its simplicity without loss of predictive capacity.

### 3.1. External Validation Analysis

Regarding the model validation, Figure 7 shows the boxplots for each real class (non-severity; severity, respectively) on the external validation dataset. The distribution of the probabilities predicted by the model for non-severity patients (green box) was clearly under the distribution of probabilities for severity patients in the red box. Thus, we observed good discrimination ability for the predictive model and a possible threshold of 0.4 that can separate the two groups.

Figure 8 shows good agreement between the predicted probabilities and real outcome in our external validation, with an intercept of −0.123 and a slope of 1.006. Moreover, the ROC curve (Figure 9, AUC = 0.821; 95% CI 0.787–0.854) demonstrates no loss in predictive ability in the external validation. The Youden index (4) was obtained with a specificity (2) and sensitivity (3) pair of 0.609 and 0.886, respectively.

The clinical utility analysis of the model is given in Table 5. For each probability threshold, the following metrics are presented: sensitivity (3), specificity (2), PPV (5), NPV (6), and accuracy (1). The cut-offs of 0.24 and 0.4 equally optimize the Youden index (4). However, if minimization of both classification errors is desired, a 0.4 threshold gives us more balanced (0.71 and 0.78) sensitivity (3) and specificity (2) values.

### 3.2. Clinically Useful Tool

The generated model could be used in clinical practice via a user-friendly web application. The necessary code to assemble the tool and apply the generated model is available at https://github.com/ITAINNOVA/covid_IIS (accessed on 6 July 2021).

The designed tool has the main functionality of predicting severity (death or ICU admission) through the best performing model given the information for the 20 explanatory variables, which are the fields to be entered into the tool. Given the patient information, the tool returns the probability of the patient ending up in a serious condition (i.e., ICU or death). Although the number of variables to be entered into the tool is not high and is consistent with practical use, the information for some of the variables may not be available for some patients. In these cases, we would encounter a problem of incompleteness, which we solve by assigning the mean value.

In addition to providing the risk given the patient′s characteristics, the interpretability and explainability of the model were explored to provide more information to the clinician. In particular, using the Shapley technique [50], the tool provides a graphical representation of the Shapley values using Python′s SHAP library [51], which allows interpretation of the direction and intensity of each variable. An example of such a representation provided by the tool for a given patient is shown in Figure 10; the patient′s age (lower than average) contributes the most (has the longest bar) to decreasing the probability of risk, but the amount of urea in the patient contributes to the contrary.

## 4. Discussion

A large number of studies have analysed predictive risk factors for COVID-19 [52,53,54], though less common are studies that have explored predictive models. Our study focused on the generation of a risk prediction model using machine learning techniques and variables that are easily obtained in the emergency room of most hospitals.

We trained neural networks, random forest and XGBoost algorithms using a hyperparameter tuning optimization. Our best model was reached using XGBoost algorithm, that taking into account the amount of candidate predictors can be a good alternative. This kind of models provides robust models as their predictions are based on an additive combination of trees that are built using different sets of data and variables. In our case, the best model was reached using 330 trees, those trees are built using the 85% of predictor variables and 64% of the training data sample, this guarantees that each tree explored the predictive ability of predictor variables in different data sample and over a different set of variables. In addition, the trees had a maximum depth of 15, preventing the overfitting that it is present in trees with too many branches.

The model we developed was built from a cohort of patients in Aragon who were hospitalized with a positive SARS-CoV-2 PCR test, and the outcome studied was ICU admission or mortality within 30 days of hospital admission. The analyses were performed for a cohort of 3623 patients, which is larger than the predictive models reviewed by Wynants et al. [23]. Several studies have analysed predictive models of mortality, but with an unclear definition of the target population, without specifying the outcome (mortality) period, and at high risk of biases [33,34,35,36]. This bias can lead to miscalibrations and overestimation of the discrimination performance, especially when they are not externally validated. Our model was validated both internally and externally to avoid overfitting of the generated model and to provide a true measure of model performance and generalizability. The internal validation was based on 12.5% of the training data and provides a good performance AUC = 0.8307 of our model. The set used for external validation maintained a proportion of patients (*n* = 626) with outcomes (death or ICU) similar to the training set (31–32%). The AUC value of 0.821 shown a minimum loss in the predictive ability of the model.

Regarding calibration, Xie et al. [39], despite achieving excellent discrimination (C index = 0.98), obtained a calibration that had a slope > 1 (probabilities too high for low-risk patients and too low for high-risk patients). Our model achieved good discrimination with the external validation set (AUC = 0.821, 95% CI 0.787–0.854) and accurate calibration (slope = 1, intercept = −0.12) [55], which means that the predicted probabilities are close to the expected probability distribution.

Most predictive models usually show their accuracy by the AUC as a measure of the discrimination ability. The Marcos et al. [28] model achieved an AUC of 0.83 (95% CI 0.81–0.85). With a similar aim, models trained by Patel et al. [30] with only the top five features achieved similar performance as those using all features; in particular, the model for ICU admission achieved an AUC of 0.79 (95% CI 0.72–0.85) and the one for mechanical ventilation achieved an AUC of 0.83 (95% CI 0.77–0.9). Knight et al. [23] evaluated the discrimination of the model, which obtained an AUC of 0.77 (95% CI 0.76–0.77), and its calibration, which was excellent (calibration-in-the-large = 0, slope = 1.0). Therefore, our model achieves a performance close to or even superior to that of the literature studies.

The study we present explored machine learning techniques for the development and validation of models with the aim of predicting an unfavourable outcome for a patient admitted with COVID-19, where an unfavourable outcome is understood to be ICU admission or mortality. In particular, MLP, random forest and XGBoost were analysed using dynamic optimization of the model′s hyperparameters. The ultimate goal of our study was to build a model that can be used in clinical practice via a user-friendly web-based tool/platform accessible by the clinician. To this end, the 20 most important variables readily available in the emergency and patient records were selected for generation of the model. The final model was built using the XGBoost algorithm and achieved an AUC of 0.821 (95% CI 0.787–0.854) in the external validation set.

In contrast to the reviewed papers, our study included a large cohort with a great amount of data. In addition, as discussed above, our study was validated both internally and externally, showing good calibration, on a set of patients that includes two different pandemic waves in the training set and a third one in the validation set, but maintaining a similar percentage of severity (unfavourable outcomes).

An analysis of the optimal cut-off points was also carried out depending on the metric to be optimized, which provides a series of thresholds for decision-making, allowing one cut-off or another to be chosen depending on the needs of the system. The probability threshold of 40% provided the best performance in a sensitivity/specificity equilibrium analysis, but we provided a clinical utility analysis on different cut-offs to choose another pair of sensitivity (3) and specificity (2) values depending on the evolution of the pandemic.

Although the generated model was externally validated with good calibration and discrimination ability, the study has some limitations. Our study was carried out with data from a single region of Spain (Aragon), but the hospitals belong to the National Health System, which has universal health coverage and, therefore, contains information from patients of different ethnicities and social characteristics. For this reason, diversity is guaranteed. Although the model considers patient variables that have been shown to be risk factors, we propose also considering the inclusion of other variables of interest in future studies, such as the type of vaccine administered or the time of vaccination. Regarding model limitations, boosting techniques are more likely to overfit than bagging although we have used hyperparameter tuning to prevent this overfitting. We also encourage validation of our model with a different cohort.

## 5. Conclusions

In this study, we incorporated information from a population that includes several pandemic waves to generate and validate a model capable of predicting death or admission to the ICU for patients admitted with COVID-19 based on 20 emergency and clinical history characteristics. The results of the internal and external validation showed a good discriminative capacity and calibration of the model, which implies that the use of the model is adequate. The final objective of the study was to design and provide an ergonomic web application (https://github.com/ITAINNOVA/covid_IIS) (accessed on 6 July 2021) that integrates the model in a way that provides the clinician with the probability that the patient admitted for COVID ends up serious, which can aid in developing an action plan.

## Figures and Tables

**Figure 1 ijerph-18-08677-f001:**
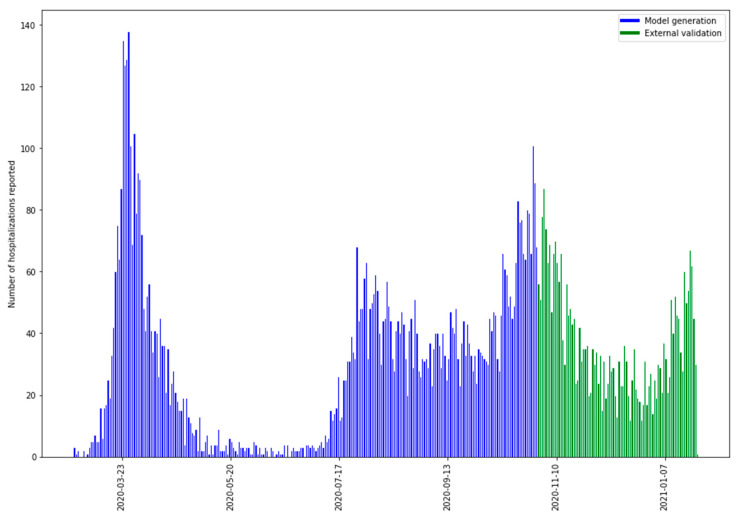
Distribution of the number of hospitalizations reported between February 2020 and January 2021.

**Figure 2 ijerph-18-08677-f002:**
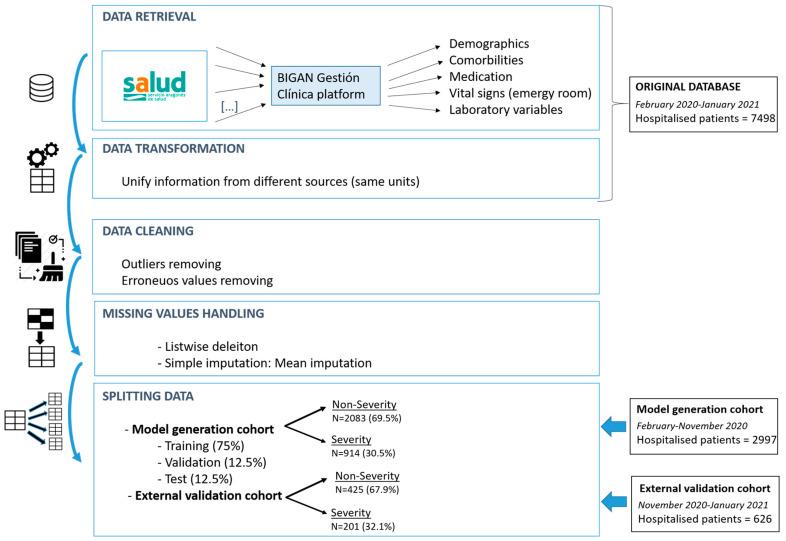
Flowchart of data preparation process.

**Figure 3 ijerph-18-08677-f003:**
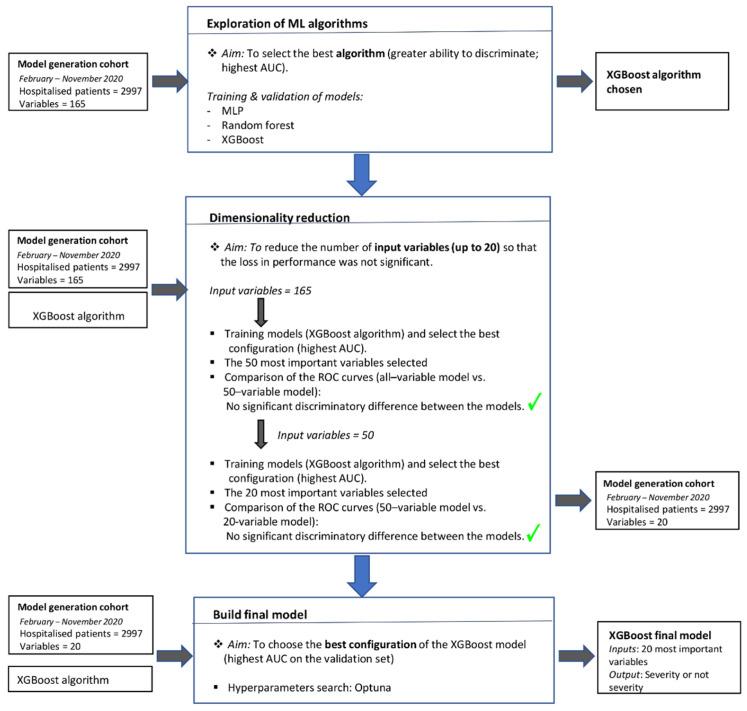
Flowchart of the final model generation process.

**Figure 4 ijerph-18-08677-f004:**
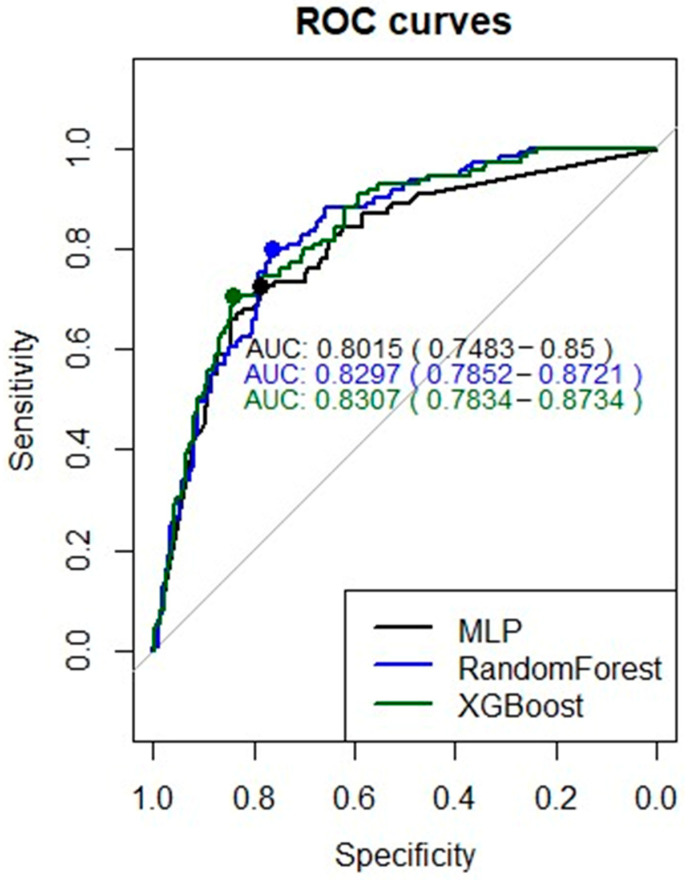
ROC curves of the best models (MLP, Random Forest, and XGBoost).

**Figure 5 ijerph-18-08677-f005:**
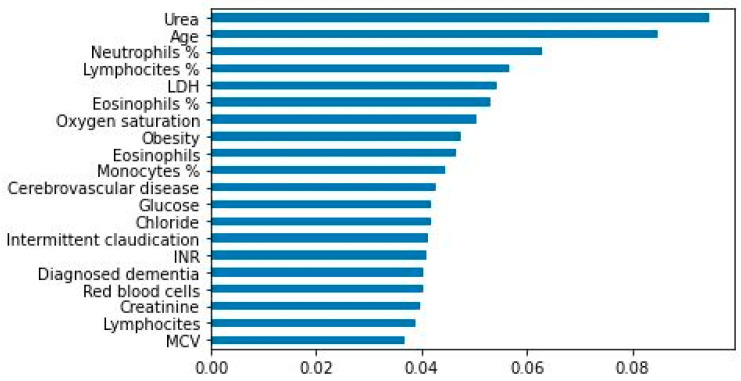
Importance diagram of the 20 variables selected.

**Figure 6 ijerph-18-08677-f006:**
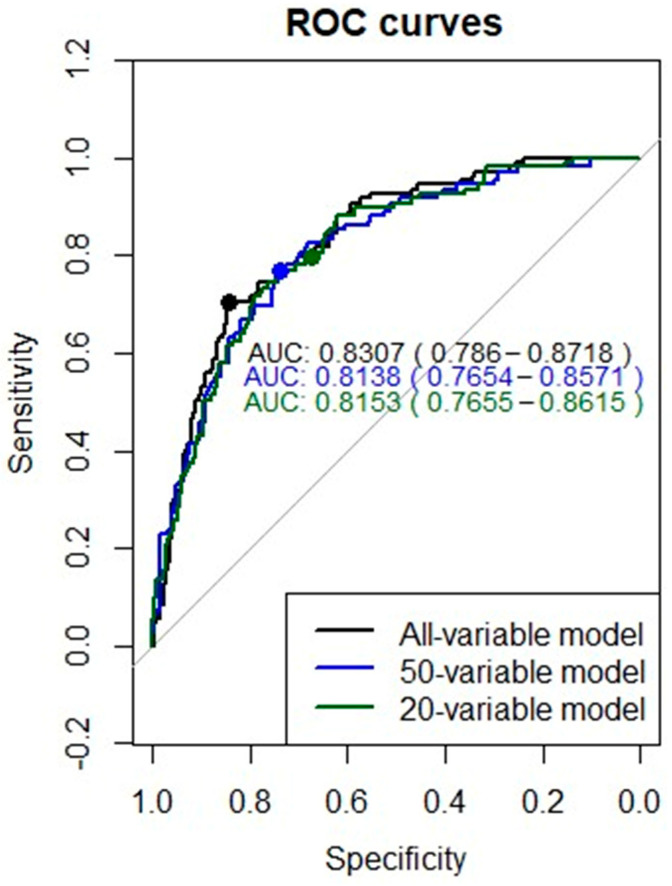
ROC curves of the analysed models considering all variables, 50 variables, and the 20 most influential variables.

**Figure 7 ijerph-18-08677-f007:**
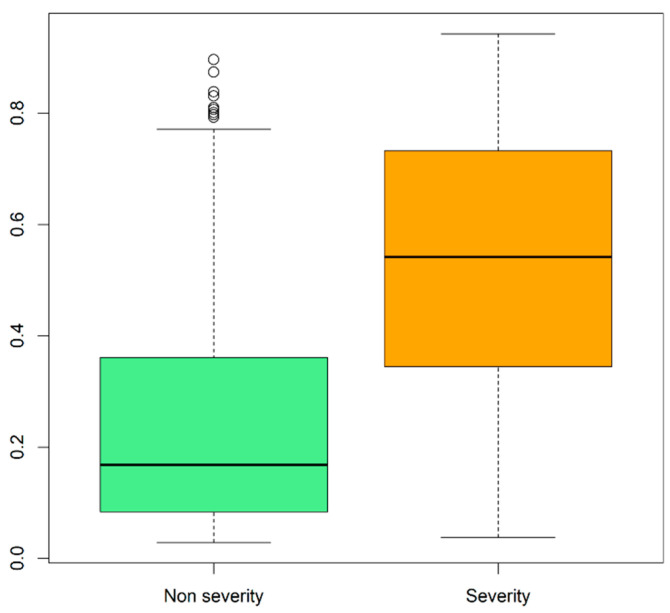
Boxplots of the probabilities predicted by the final model for the external validation set grouped by the actual class group.

**Figure 8 ijerph-18-08677-f008:**
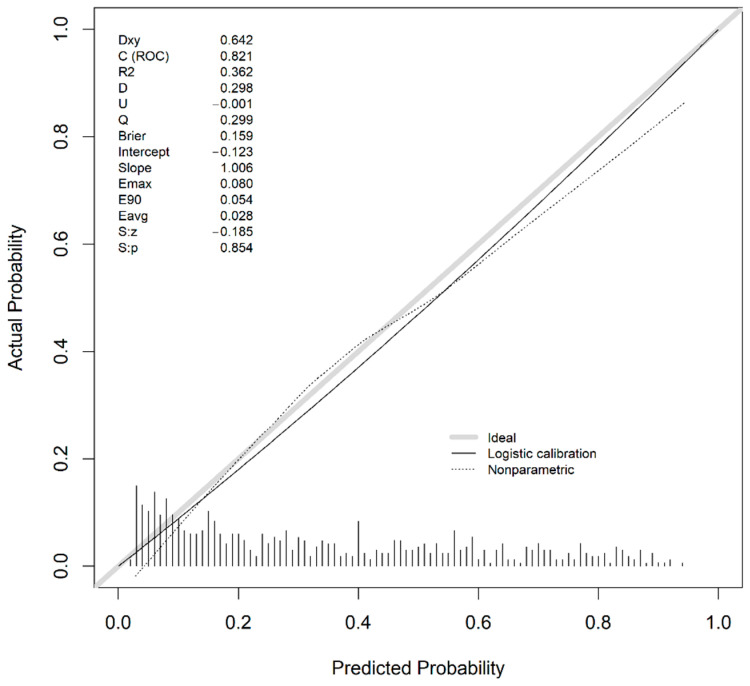
Calibration curve for the external validation dataset.

**Figure 9 ijerph-18-08677-f009:**
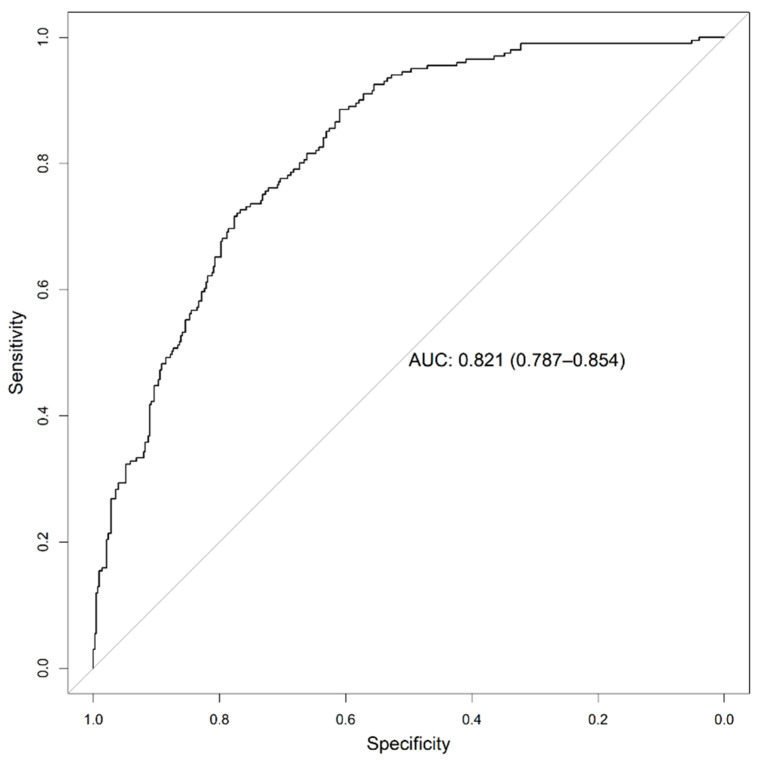
ROC curve for the external validation dataset.

**Figure 10 ijerph-18-08677-f010:**
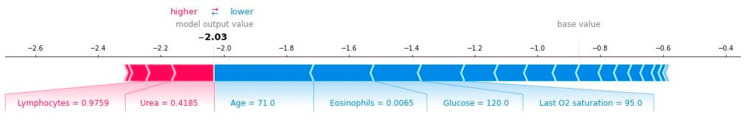
Example of a graphical representation of the Shapley values provided by the tool. *Base value*: mean prediction of the test set. Blue variables contribute to decreasing the probability of the patient risk, whereas red variables contribute to increasing the probability of patient risk. The magnitude of such increase or decrease is measured by the length of the bar.

**Table 1 ijerph-18-08677-t001:** Demographic and clinical information of patients with and without severe disease in the model generation cohort (February–November 2020).

	Non Severity (*n* = 1548)	Severity (*n* = 699)	*p*-Value
Age (years)	66 (51–81)	83 (71–89)	<0.001
Oxygen saturation (%)	96 (94–97)	94 (91–96)	<0.001
Intermittent claudication (yes)	41 (2.65%)	42 (6.01%)	<0.001
Cerebrovascular disease (yes)	104 (6.72%)	104 (14.88%)	<0.001
Dementia (yes)	124 (8.01%)	141 (20.17%)	<0.001
Obesity (yes)	218 (14.08%)	123 (17.6%)	0.03701
Chloride (mmol/L)	101.5 (98.4–104)	102.3 (99–106)	<0.001
Creatinine (mg/dL)	0.86 (0.68–1.1)	1.1 (0.81–1.58)	<0.001
Eosinophils (%)	0.17 (0–0.7)	0 (0–0.2)	<0.001
Eosinophils (mil/mm^3^)	0.0102 (0–0.04385)	0 (0–0.01702)	<0.001
Glucose (mg/dL)	110 (96–133)	128 (104–170)	<0.001
International normalized ratio-prothrombin time (INR-PT)	1.1 (1.02–1.17)	1.16 (1.06–1.305)	<0.001
Lactate dehydrogenase (U/L)	275 (219–350)	336.5 (246.8–470)	<0.001
Lymphocytes (%)	17.9 (11.65–25.98)	10.74 (6.275–18)	<0.001
Lymphocytes (mil/mm^3^)	1.0875 (0.7561–1.4941)	0.8055 (0.5605–1.1458)	<0.001
Monocytes (%)	8.07 (6–10.438)	6.2 (4–8.9)	<0.001
Neutrophils (%)	72 (63.28–80.41)	81.4 (72.75–88.5)	<0.001
Red blood cells (mil/mm^3^)	4.6 (4.19–4.95)	4.32 (3.87–4.72)	<0.001
Urea (g/l)	0.3595 (0.27–0.5258)	0.609 (0.42–0.91)	<0.001
Mean corpuscular volume (fl)	89.7 (86.1–93)	91.5 (87.7–95)	<0.001

Data are presented as *n* (%) or median (interquartile range).

**Table 2 ijerph-18-08677-t002:** Demographic and clinical information of patients with and without severe disease in the external validation cohort (November 2020–January 2021).

	Non Severity (*n* = 425)	Severity (*n* = 201)	*p*-Value
Age (years)	71 (58–83)	84 (74–89)	<0.001
Oxygen saturation (%)	95 (94–97)	95 (92–97)	0.03714
Intermittent claudication (yes)	24 (5.65%)	17 (8.46%)	0.2484
Cerebrovascular disease (yes)	44 (10.35%)	37 (18.41%)	0.007451
Dementia (yes)	43 (10.12%)	32 (15.92%)	0.05051
Obesity (yes)	75 (17.65%)	27 (13.43%)	0.2236
Chloride (mmol/L)	101.7 (98.9–104.1)	102.1 (98.3–105.7)	0.1273
Creatinine (mg/dL)	0.85 (0.68–1.06)	1.14 (0.8–1.64)	<0.001
Eosinophils (%)	0.1 (0–0.4)	0.01 (0–0.24)	<0.001
Eosinophils (mil/mm^3^)	0.00715 (0–0.02808)	0.00026 (0–0.0162)	<0.001
Glucose (mg/dL)	116 (100–145)	130 (106–174)	<0.001
International normalized ratio-prothrombin time (INR-PT)	1.08 (1.02–1.17)	1.15 (1.06–1.32)	<0.001
Lactate dehydrogenase (U/L)	266 (213–336)	322 (245–414)	<0.001
Lymphocytes (%)	16 (10.5–24.4)	9.2 (5.5–14.8)	<0.001
Lymphocytes (mil/mm^3^)	0.9432 (0.6696–1.362)	0.6987 (0.4611–1.0577)	<0.001
Monocytes (%)	7.9 (5.6–10.59)	5.7 (3.6–8.29)	<0.001
Neutrophils (%)	74.7 (64.66–82.4)	84.5 (75.1–88.7)	<0.001
Red blood cells (mil/mm^3^)	4.46 (4.01–4.81)	4.17 (3.66–4.62)	<0.001
Urea (g/L)	0.396 (0.306–0.553)	0.672 (0.446–1.1)	<0.001
Mean corpuscular volume (fl)	90.1 (87.1–93)	90.7 (87–94.9)	0.07457

**Table 3 ijerph-18-08677-t003:** Search space of the hyperparameters explored for each algorithm.

Model	Parameters	Search Space	Best Model	AUC
	Number of hidden layers	[2, 10]	7	
	Number of neurons	[16, 512]	[96, 176, 240, 240, 352, 352, 384]	
MLP	Activation layer	[selu, linear, tanh, softmax]	[softmax, selu, softmax, selu, selu, selu, selu]	0.8015
	Learning rate	{0.001,0.01,0.1}	0.001	
	Optimizer	{sgd, adam, rmsprop}	rmsprop	
	Batch size	[1, 64]	54	
	Number of estimators	[30, 1300]	820	
	Max. depth	[3, 20]	15	
Random Forest	Min. samples split	[2, 30]	10	0.8297
	Criterion	{gini, entropy}	gini	
	Min. impurity decrease	{5 × 10^−5^, 1 × 10^−4^, 2 × 10^−4^, 5 × 10^−4^, 1 × 10^−3^, 1.5 × 10^−3^, 2 × 10^−3^, 5 × 10^−3^, 0.01}	2 × 10^−4^	
	Number of estimators	[30, 1300]	330	
	Scale pos. weight	[1, 10]	1	
	Column subsample size per tree	[0.3, 1]	0.85	
	Subsample size per tree	[0.3, 1]	0.64	
XGBoost	Max. depth	[3, 20]	15	0.8307
	Learning rate	{10^−4^, 10^−3^, 10^−2^,0.1, 0.15, 0.2, 0.3, 0.4}	0.01	
	Reg. alpha	{10^−4^, 10^−3^, 10^−2^,0.1, 0.15, 0.2, 0.3, 0.4}	0.4	
	Gamma	[0.05, 1]	0.6	

Sgd: stochastic gradient descendent; Max. depth: The maximum tree depth for the base learners; Min. samples split: Minimum number of samples remaining in a node to consider splitting it; Min. impurity decrease: A node will be split if this split induces a decrease in the impurity greater than or equal to this value; Scale pos. weight: The balance between positive and negative classes; Reg. alpha: The L1 regularization on the weights; Gamma: The minimum loss reduction required to further partition a leaf node of the tree. Column subsample size per tree and Subsample size per tree describe the ratio of columns and rows, respectively, used in each boosting round, and Learning rate is the boosting learning rate.

**Table 4 ijerph-18-08677-t004:** Hyperparameter configuration of the best XGBoost models.

Parameter	All-Variable Model	50-Variable Model	20-Variable Model
Number of estimators	330	690	340
Scale pos. weight	1	5	1
Column subsample size per tree	0.85	0.41	0.84
Subsample size per tree	0.64	0.94	0.68
Max. depth	15	14	13
Learning rate	0.01	0.2	0.01
Reg. alpha	0.4	0.3	0.15
Gamma	0.6	0.1	0.45

**Table 5 ijerph-18-08677-t005:** Cut-off analysis of the external validation cohort (November 2020–January 2021).

Thr	tp	tn	fp	fn	Sens	Spec	PPV	NPV	Accuracy	Youden
0.05	199	44	381	2	0.99	0.1	0.34	0.96	0.39	0.09
0.1	199	137	288	2	0.99	0.32	0.41	0.99	0.54	0.31
0.15	192	187	238	9	0.96	0.44	0.45	0.95	0.61	0.4
0.2	184	237	188	17	0.92	0.56	0.49	0.93	0.67	0.47
0.22	179	250	175	22	0.89	0.59	0.51	0.92	0.69	0.48
0.24	178	257	168	23	0.89	0.6	0.51	0.92	0.69	0.49
0.26	171	267	158	30	0.85	0.63	0.52	0.9	0.7	0.48
0.28	164	277	148	37	0.82	0.65	0.53	0.88	0.7	0.47
0.3	159	288	137	42	0.79	0.68	0.54	0.87	0.71	0.47
0.32	154	300	125	47	0.77	0.71	0.55	0.86	0.73	0.47
0.34	152	307	118	49	0.76	0.72	0.56	0.86	0.73	0.48
0.36	148	318	107	53	0.74	0.75	0.58	0.86	0.74	0.48
0.38	146	326	99	55	0.73	0.77	0.6	0.86	0.75	0.49
0.4	143	330	95	58	0.71	0.78	0.6	0.85	0.76	0.49
0.42	134	339	86	67	0.67	0.8	0.61	0.83	0.76	0.46
0.44	131	343	82	70	0.65	0.81	0.62	0.83	0.76	0.46
0.46	125	345	80	76	0.62	0.81	0.61	0.82	0.75	0.43
0.48	117	353	72	84	0.58	0.83	0.62	0.81	0.75	0.41
0.5	113	359	66	88	0.56	0.84	0.63	0.8	0.75	0.41

Thr: Probability threshold, tp: True positive, tn: True negative, fp: False positive, fn: False negative, Sens: Sensitivity = tp/(tp + fn), Spec: Specificity = tn/(tn + fp), PPV: Positive predictive value = tp/(tp + fp), NPV: Negative predictive value = tn/(tn + fn), Accuracy = (tp + tn)/(tp + fp + tn + fn), Youden = Sens + Spec − 1 = tp/(tp + fn) + tn/(tn + fp) − 1.

## Data Availability

Data sharing not applicable.

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
