# Peer review of "A Clinical Decision Web to Predict ICU Admission or Death for Patients Hospitalised with COVID-19 Using Machine Learning Algorithms"

_ijerph, 2021, doi:10.3390/ijerph18168677_

Round 1

Reviewer 1 Report

This manuscript presents the methodology for the clinical decision web to predict ICU admission or death for patients hospitalised with COVID-19 using ML algorithms. Even though the paper is interesting, it suffers from several stylistic and methodological shortcomings that need to be modified to bring the article to a level worthy of publication in a journal of this rank.

There are some concerns:

In my point opinion, this paper demonstrates a very interesting application of artificial intelligence algorithms in order to predict the outcome of the patient hospitalised with COVID-19. However, in the Introduction, the novelty of the paper should be outlined better.

 Furthermore, as the importance of the topic, I recommend updating the literature, by including the papers recently reported regarding the application of AI in various fields of COVID-19 problem-solving strategies (for your convenience:  “doi.org/10.1177/03000605211000157 ”, “doi.org/10.3390/ijerph18084287“,  “doi.org/10.1186/s12911-021-01488-9 ”, “doi.org/10.4108/eai.7-7-2021.170287” )

The sentence in line 91 „ Missing data were imputed as the mean value. “ should be commented in more detail.

The content of this paper is well structured, English needs a revision for minor spelling/grammar errors.

Add a legend of abbreviations used.

Reviewer 2 Report

From my point of view would be interesting to have a look in:

Introduction

In line 43 COVID-19 you need define this initial.

  1. Materials and Methods

In line 72 RT-PCR -> initial

I believe all initials that are mentioned the first time needs to be defined. The readers need to understand it.

In line 101, please explain the external validation with a brief explanation.

In 2.3. Training and validation was chosen XGBoost algorithm, in this case to best reading a block diagram will clarify the main idea.

Analyses were per- 163 formed using R language, would be nice to have block diagram ou algorithm to this script.

In figure 3 I believe AUC = 0.831 means 0.8297 (truncated) In this case you need give more details and explain why is better.

In figure 5 when the author say, best model, are the authors referring about AUc number, this is not clear in the manuscript.

In line 225 there is no table 7, just Figure 7.

In table 5 sensitivity, specificity, PPV, NPV, and accuracy and sensitivity, specificity, PPV, NPV, and accuracy needs to be defined before.

Discussion When you an using ML, AI the authors needs to give more detail, like if you are using Neural Network is possible to indicate how may layers, neurons etc…

Line 294 “Although this type of validation is adequate”, here is an example that needs one metric to validate (error)

Conclusion:

In introduction the authors comments “model was 30 validated both internally and externally” but in conclusion was not mentioned about internally.

“The results of the external validation showed a good discriminative 389 capacity and calibration of the model, which implies that the use of the model is adequate.”

Here I missing some metric to say “good or not”, in manuscript I did not see how can I reach that. In de model seems to me very superficial, so need to give mode information about de model, using block diagram etc.. Just to refer other manuscript take me more time to understand and check it.

Reviewer 3 Report

This work is mainly about the condition prediction of patients hospitalized with COVID-19 by using machine learning methods. The 3623 patients 23 with confirmed COVID-19 from 23 hospitals, between February 2020 and January 2021, (SSA) are used for case study. Three machine learning (ML) algorithms, including multilayer perceptron, random forest, and extreme gradient boosting (XGBoost) were explored. Sufficient experiments are used to verify the effectiveness of the designed methods and decision web. Overall, the procedure and framework of this manuscript are clear. It is a good work. However, before acceptance, several major comments should be addressed:

  1. The motivations and contributions should be added and listed point by point in the intrdocution. Besides, the authors may give brief descriptions about the used machine learning algorithms. The descriptions about your background and used data have been sufficent.
  2. In discussion part, the related works were reviewed. However, it looks strange here. The literature review should be written in Introduction part. And then highlight your motivations and contributions. The discussion part should focus on the analysis and discussions about the achieved results, instead of literature review..
  3. This work used the shallow machine learning methods, including MLP, random forest and XGBoost. Since the deep learning models have been emerging across different fields, the authors need to give a more comprehensive review about the works using deep learning in Introduction. The reviewer recommended to add more discussion on this point, the differences between shallow machine learning methods and deep learning methods. The authors may refer to: "A hybrid generalization network for intelligent fault diagnosis of rotating machinery under unseen working conditions," in IEEE Transactions on Instrumentation and Measurement, doi: 10.1109/TIM.2021.3088489.
  4. In figure 9, what is the mearning of magnitude. May be the influence of each variable? The authors may use the age as an example. Once the age is lower than 71, the risk will be decreased. And the decrease degree is represented by the length of blue part. Is it right?
  5. The purpose of the study was to build a predictive model for estimating the risk of ICU 21 admission or mortality. The authors adopted the ROC, boxplots to show the predictive results. However, for a clinicians, it will be more clear to see the diagnosis results by your predictive model, such as severity or not severity. Consequenlty, a specific diagnosis results should be reported along with the current representations. It is a classification problem. The confusion matrix and classification (diagnosis) accuracy may be a good option to improve clinical decision web.

Reviewer 4 Report

Thank you Editor for invitation me to review this interesting manuscript.

I enjoy to review this manuscript.

Following comments should be taken into account:

  1. Please add graphical abstract. This will help authors to introduce their paper in the better manner.
  2. Grammatical issues in the manuscript should be corrected
  3. Comparison between the current work and previous works, especially those ones used artificial intelligence, is highly recommended.
  4. The legend of figures should enlarge.
  5. Please add one section about limitation of your ML model
  6. The author should use the following papers in the introduction section for referring the readers to other applications of neural network in different sciences such as CO2 storage and mitigation climate change
  7. Application of artificial neural network for predicting the performance of CO2 enhanced oil recovery and storage in residual oil zones. https://doi.org/10.1038/s41598-020-73931-2
  8. I would suggest comparing your work with the previous studies, including application of different AI techniques in terms of errors and efficiency.
  9. I would recommend depicting the effect of boundaries on the performance of the Machine Learning model.

Round 2

Reviewer 2 Report

In equations in page 9, all needs to be enumerated and give references each on.

In Table 1, give a look, some text out of line.

Author Response

We thank the reviewer for the review and comments. All have been addressed and have been highlighted in green in the document:

  • In equations in page 9, all needs to be enumerated and give references each on.
  • In Table 1, give a look, some text out of line.

The equations have been numbered and are referenced throughout the text by their corresponding number. The text in table 1 has also been corrected as suggested by the reviewer.

Reviewer 3 Report

The authors have addressed all my comments. The reivewer would like to recommend the "Acceptance".

Author Response

We thank the reviewer for his/her review and recommendation.